# Genome-Wide Association Study Identifies Genetic Polymorphisms Associated with Estimated Minimum Effective Concentration of Fentanyl in Patients Undergoing Laparoscopic-Assisted Colectomy

**DOI:** 10.3390/ijms24098421

**Published:** 2023-05-08

**Authors:** Daisuke Nishizawa, Tsutomu Mieda, Miki Tsujita, Hideyuki Nakagawa, Shigeki Yamaguchi, Shinya Kasai, Junko Hasegawa, Kyoko Nakayama, Yuko Ebata, Akira Kitamura, Hirotomo Shimizu, Tadayuki Takashima, Masakazu Hayashida, Kazutaka Ikeda

**Affiliations:** 1Addictive Substance Project, Tokyo Metropolitan Institute of Medical Science, Tokyo 156-8506, Japan; nishizawa-ds@igakuken.or.jp (D.N.);; 2Department of Anesthesiology, Saitama Medical University Hospital, Saitama 350-0495, Japan; 3Department of Anesthesiology, Saitama Medical University International Medical Center, Saitama 350-1298, Japan; 4Division of Colorectal Surgery, Department of Surgery, Tokyo Women’s Medical University, Tokyo 162-8666, Japan; 5Laboratory for Safety Assessment and ADME, Asahi Kasei Pharma Corporation, Shizuoka 410-2321, Japan; shimizu.hh@om.asahi-kasei.co.jp (H.S.); takashima.tn@om.asahi-kasei.co.jp (T.T.); 6Department of Anesthesiology and Pain Medicine, Juntendo University School of Medicine, Tokyo 113-8421, Japan

**Keywords:** opioids, analgesics, single-nucleotide polymorphism, genome-wide association study, minimum effective concentration, laparoscopic-assisted colectomy

## Abstract

Sensitivity to opioids varies widely among individuals. To identify potential candidate single-nucleotide polymorphisms (SNPs) that may significantly contribute to individual differences in the minimum effective concentration (MEC) of an opioid, fentanyl, we conducted a three-stage genome-wide association study (GWAS) using whole-genome genotyping arrays in 350 patients who underwent laparoscopic-assisted colectomy. To estimate the MEC of fentanyl, plasma and effect-site concentrations of fentanyl over the 24 h postoperative period were estimated with a pharmacokinetic simulation model based on initial bolus doses and subsequent patient-controlled analgesia doses of fentanyl. Plasma and effect-site MECs of fentanyl were indicated by fentanyl concentrations, estimated immediately before each patient-controlled analgesia dose. The GWAS revealed that an intergenic SNP, rs966775, that mapped to 5p13 had significant associations with the plasma MEC averaged over the 6 h postoperative period and the effect-site MEC averaged over the 12 h postoperative period. The minor G allele of rs966775 was associated with increases in these MECs of fentanyl. The nearest protein-coding gene around this SNP was *DRD1*, encoding the dopamine D_1_ receptor. In the gene-based analysis, the association was significant for the *SERP2* gene in the dominant model. Our findings provide valuable information for personalized pain treatment after laparoscopic-assisted colectomy.

## 1. Introduction

Opioids, such as morphine, fentanyl, oxycodone, and hydromorphone, are widely used as effective analgesics for the treatment of acute and chronic pain because of their robust antinociceptive effects. However, effects of opioids are not uniform across all patients, and considerable differences in the responsiveness or sensitivity to opioids are widely known [1,2]. This can influence the total amount of analgesics that are required for adequate pain relief, which can hamper the effective treatment of pain in clinical practice. For example, the minimum effective concentrations (MECs) of fentanyl at which patients demand additional fentanyl doses to relieve recurring postoperative pain are reported to vary widely among patients from 0.23 to 0.99 ng/mL after orthopedic surgery [3], from 0.23 to 1.18 ng/mL after open abdominal surgery [4], from 0.30 to 1.45 (5–95 percentiles) ng/mL after open abdominal surgery [5], and from 0.2 to 8.0 ng/mL after various surgical procedures [6], indicating that MECs of fentanyl after certain surgical procedures have a more than four- to fivefold difference among individuals [7]. Because of such significant differences in opioid sensitivity, empirical methods of administration that have been utilized by trial and error are an imperfect practice that can result in delayed or inadequate analgesia and possibly overdose [7].

The required amounts of opioid analgesics may also vary among patients with pain depending on age, sex, weight, basal pain sensitivity, the type of surgery, perceived pain during the perioperative period [2], and genetic factors. Previous twin studies of experimental heat and cold pressor pain reported that genetic effects were estimated to account for 12%, 60%, and 30% of the observed response variance (i.e., pain threshold) after administration of the opioid analgesic alfentanil for heat pain, cold-pressor pain, and cold-pressor pain, respectively [8,9]. Although the variance of responses to opioids appears to be moderately influenced by genetic factors, potential genes and genetic variants that are involved in response variance have not yet been fully elucidated. Further studies are needed to delineate such genetic factors.

To date, many candidate gene association studies have been conducted [10,11,12]. These studies have targeted various genes that are involved in pharmacokinetic and pharmacodynamic opioidergic pathways and pain-related genes of various modalities, such as the μ-opioid receptor (*OPRM1*) gene; cytochrome P450, family 2, subfamily D, polypeptide 6 (*CYP2D6*) gene; adenosine triphosphate-binding cassette (ABC), subfamily B (MDR/TAP), member 1 (*ABCB1*) gene; catechol-*O*-methyltransferase (*COMT*) gene; and genes that are related to cytokines (e.g., interleukin-1β, interleukin-6, and tumor necrosis factor-α) [2]. Additional candidate genes are detailed in previous review articles [10,11,12]. Genetic factors that are related to opioid sensitivity and responsiveness can also be explored using a genome-wide approach in genome-wide association studies (GWASs), although only a few studies have conducted GWASs of such phenotypes. One example is a prospective cross-sectional multinational multicenter study of patients with cancer from 11 European countries [13] who were treated with opioids for moderate or severe pain. The strongest association with responsiveness to opioids was found for the rs12948783 single-nucleotide polymorphism (SNP), which is located upstream of the *RHBDF2* gene [14].

We also conducted GWASs of phenotypes that are related to opioid sensitivity and candidate gene studies [15,16,17,18,19,20,21,22,23,24,25,26,27,28,29,30,31,32,33]. In our GWASs, although genetic variants that were significantly associated with opioid responsiveness for the treatment were not found in patients with chronic pain [30], we identified several SNPs, including the rs2952768 SNP (located near the *METTL21A* [*FAM119A*] and *CREB1* gene regions), that were significantly associated with postoperative opioid analgesic requirements in subjects who underwent cosmetic orthognathic surgery for mandibular prognathism [18]. Furthermore, a GWAS of patients who were treated with opioid analgesics for the treatment of cancer pain identified several SNPs that were significantly associated with average daily opioid requirements for the treatment of pain, the best candidates of which were the rs1283671 and rs1283720 SNPs in the *ANGPT1* gene region. We also conducted GWASs of subjects who underwent laparoscopic-assisted colectomy (LAC), a surgery that is often categorized as minimally invasive because of much smaller skin incisions and less postoperative pain compared with traditional open abdominal surgery, although postoperative pain is not “minimal” after surgery [22]. Our GWASs of subjects who underwent LAC identified several potent SNPs, including the nonsynonymous rs2076222 SNP in the *LAMB3* gene region, the rs199670311 nonsynonymous SNP in the *TMEM8A* gene region, and intronic SNPs, including rs4839603, in the *SLC9A9* gene region [22,25].

Likely because of relative facileness, most previous human genetic studies have focused on opioid analgesic requirements for the treatment of disease-related pain, chronic pain, and perioperative/postoperative pain as the main endpoint to investigate genetic variants that are associated with human responsiveness and sensitivity to opioids [10,11,12,13,14,15,16,17,18,19,20,21,22,23,24,25,26,27,28,29,30,31,32,33]. However, MECs at the plasma and effect site are not generally measured directly. Thus, these parameters have not been used to date in human genetic association studies. Nevertheless, with the development of pharmacokinetic/pharmacodynamic knowledge and the advancement of computer technology, it has become easier to simulate the process of plasma or effect-site concentrations of anesthetics and analgesics by leveraging related simulation software programs, such as STANPUMP (http://opentci.org/code/stanpump; accessed on 25 January 2023) and tivatrainer (https://www.tivatrainer.com; accessed on 25 January 2023). Such simulation software has been used in many studies to estimate plasma and effect-site concentrations of anesthetics and analgesics [34,35,36,37,38,39].

In the present study, we conducted a GWAS of patients who underwent LAC to identify potential genetic variants that contribute to the efficacy of opioid analgesics based on information about estimated plasma or effect-site concentrations of fentanyl, which were calculated by utilizing one of the programs, the BeConSim Monitoring simulation software program (http://www.masuinet.com; accessed on 1 January 2020) [40,41,42].

## 2. Results

### 2.1. Impact of Clinical Variables on Estimated MEC of Fentanyl in Subjects Who Underwent LAC

All 351 subjects completed the study. However, data were incomplete for one subject, particularly postoperative data. Therefore, postoperative data from the remaining 350 subjects were analyzed. Demographic, anesthetic, and surgical data for all 351 subjects are detailed in Appendix A and our previous reports [22,25].

Spearman’s rank correlation analysis indicated significant correlations among the 0–6 h plasma MEC, 0–12 h plasma MEC, 0–6 h effect-site MEC, and 0–12 h effect-site MEC (*ρ* = 0.976, *p* = 1.042 × 10^−231^, between 0–6 h plasma MEC and 0–12 h plasma MEC; *ρ* = 0.994, *p* < 1 × 10^−307^, between 0–6 h plasma MEC and 0–6 h effect-site MEC; *ρ* = 0.975, *p* = 7.383 × 10^−228^, between 0–6 h plasma MEC and 0–12 h effect-site MEC; *ρ* = 0.968, *p* = 1.043 × 10^−211^, between 0–12 h plasma MEC and 0–6 h effect-site MEC; *ρ* = 0.993, *p* < 1 × 10^−307^, between 0–12 h plasma MEC and 0–12 h effect-site MEC; *ρ* = 0.979, *p* = 3.049 × 10^−242^, between 0–6 h effect-site MEC and 0–12 h effect-site MEC). The Mann–Whitney test revealed no significant difference in the 0–6 h plasma MEC between sites of resection (*p* = 0.780), between anatomical extents of lymph node dissection (*p* = 0.740), or between genders (*p* = 0.177). The 0–12 h plasma MEC, 0–6 h effect-site MEC, and 0–12 h effect-site MEC were not significantly different between these parameters (details not shown). Multiple linear regression analyses revealed that the log-transformed 0–6 h plasma MEC was significantly associated with several clinical parameters, such as age (*β* = 0.004, *p* = 1.905 × 10^−3^), the average remifentanil infusion rate (*β* = 0.409, *p* = 1.454 × 10^−2^), the dose of fentanyl given around the end of surgery (*β* = 0.001, *p* = 4.400 × 10^−20^), and the 2 h postoperative pain score (*β* = 0.019, *p* = 1.415 × 10^−3^), and the trend was similar for 0–12 h plasma, 0–6 h effect-site, and 0–12 h effect-site MECs (details not shown). Therefore, these clinical variables were used as covariates in the subsequent analyses in the association study. Despite the strong correlations among the major endpoint variables, the 0–6 h plasma MEC, 0–12 h plasma MEC, 0–6 h effect-site MEC, and 0–12 h effect-site MEC, GWASs were performed for all of these four phenotypes in case even slight differences in these endpoint values could be caused by some slightly or moderately different genetic variants.

### 2.2. Identification of Genetic Polymorphisms Associated with Estimated MEC of Fentanyl in Patients Who Underwent LAC by GWAS

We then explored the association between genetic variations and opioid sensitivity, which was evaluated as the estimated plasma and effect-site MECs after surgery in a total of 350 subjects who underwent LAC that involved the administration of opioid analgesics [22,25]. The surgical procedure was relatively uniform; thus, invasiveness and the resultant pain were regarded as relatively homogeneous among subjects. GWASs were conducted as a consecutive three-stage analysis to identify potent SNPs that were associated with the estimated 0–6 h plasma MEC, 0–12 h plasma MEC, 0–6 h effect-site MEC, and 0–12 h effect-site MEC. Consequently, 14, 26, and 10 SNPs were selected as the top candidates in the additive, dominant, and recessive models, respectively, after the final stage for the 0–6 h plasma MEC (Appendix A). For the 0–12 h effect-site MEC, 14, 28, and 8 SNPs were selected as the top candidates in the additive, dominant, and recessive models, respectively, after the final stage (Appendix A). Similarly, 21, 24, and 19 SNPs were initially selected in the additive, dominant, and recessive models, respectively, for the 0–12 h plasma MEC. Likewise, 17, 25, and 10 SNPs were initially selected in the additive, dominant, and recessive models, respectively, for the 0–6 h effect-site MEC (details not shown). The potent SNP lists are presented in Table 1 and Table 2 and Appendix A. Among these, one SNP, rs966775, that mapped to 5p13 (GRCh37) showed significant associations with the 0–6 h plasma MEC and 0–12 h effect-site MEC after the final stage in the additive model (combined *β* = 0.0916, nominal *p* = 1.027 × 10^−7^, for the 0–6 h plasma MEC; combined *β* = 0.1071, nominal *p* = 1.299 × 10^−7^, for the 0–12 h effect-site MEC; Table 1 and Table 2). The observed *p* values for this SNP, calculated as −log10 (*p* value), deviated from the expected values from the null hypothesis of uniform distribution in the quantile–quantile (QQ) *p*-value plots for the entire sample (Appendix A for the 0–6 h plasma MEC, Appendix A for the 0–12 h effect-site MEC). Similar strong associations with this SNP were observed for the 0–12 h plasma MEC and 0–6 h effect-site MEC after the final stage in the additive model (combined *β* = 0.0908, nominal *p* = 1.206 × 10^−7^, for the 0–12 h plasma MEC; combined *β* = 0.1095, nominal *p* = 1.942 × 10^−7^, for the 0–6 h effect-site MEC; Appendix A), although the associations were not significant. The rs966775 SNP is located in the intergenic region, and the nearest protein-coding gene from this SNP position was *DRD1*, which encodes the dopamine D_1_ receptor (Appendix A). A linkage disequilibrium (LD) block that includes the rs966775 SNP was assumed to span the approximately 1 kbp chromosomal region, and no SNPs showed high LD with this SNP in the neighboring region, including the *DRD1* gene region (pairwise calculated *r*^2^ = 0.93; Appendix A). When MECs (in ng/mL) were log-transformed and shown as mean ± standard error of the mean (SEM), 0–6 h plasma MECs were 0.4960 ± 0.0171, 0.5289 ± 0.0192, and 0.6839 ± 0.0393, and 0–12 h effect-site MECs were 0.5393 ± 0.0188, 0.5839 ± 0.0220, and 0.7589 ± 0.0433, in subjects with the A/A (*n* = 171), A/G (*n* = 137), and G/G (*n* = 41) genotypes of this SNP, respectively. The copy number of the minor G allele was associated with higher 0–6 h plasma and 0–12 h effect-site MECs. A similar trend was observed for 0–12 h plasma and 0–6 h effect-site MECs, and the copy number of the minor G allele was associated with greater MEC values for these phenotypes. The genotype distribution of this SNP met the criteria of the Hardy–Weinberg equilibrium tests (*χ^2^* = 2.7283, *p* = 0.0986).

### 2.3. Identification of Genes and Gene Sets Associated with Estimated MEC of Fentanyl in Patients Who Underwent LAC by Gene-Based and Gene-Set Analyses

Considering that effects of individual markers tend to be too weak to be detected by comprehensive analyses, such as GWASs, which target only single polymorphisms, we conducted gene-based and gene-set analyses, which are statistical methods that are used to analyze multiple genetic markers simultaneously to determine their joint effect. In both analyses using MAGMA software [43], which was made accessible in the FUMA GWAS platform [44], we investigated genes and gene sets that were related to the estimated MEC of fentanyl in a total of 350 patients who underwent LAC. As a result, 921,239 SNPs from the selected candidate genes and gene sets in the additive, dominant, and recessive models were included in the analyses of all patients. The top 20 candidate genes that were found in each genetic model by the gene-based analysis are listed in Table 3. In the dominant model, *SERP2*, the top candidate gene, was significantly associated with the 0–6 h plasma MEC (adjusted *p* = 0.02425; Table 3, Figure 1B), 0–12 h plasma MEC (adjusted *p* = 0.02438; Appendix A), and 0–12 h effect-site MEC (adjusted *p* = 0.03635; Table 4, Figure 2B). The association between the *SERP2* gene and the 0–6 h effect-site MEC was marginally significant (adjusted *p* = 0.05245; Appendix A). However, in both the additive and recessive models, none of the genes were significantly associated with the phenotypes (Table 3 and Table 4; Appendix A; and Figure 1A,C and Figure 2A,C). The top 20 candidate gene sets for each phenotype that were found in each genetic model by the gene-set analysis are listed in Appendix A. As a result, the 0–6 h plasma MEC was significantly associated with the “go_paracrine_signaling” (adjusted *p* = 0.01093), “reactome_free_fatty_acid_receptors” (adjusted *p* = 0.01430), and “go_taste_receptor_activity” (adjusted *p* = 0.03004) gene sets in the additive model (Appendix A) and the “go_negative_regulation_of_epidermal_cell_differentiation” (adjusted *p* = 0.01728), “go_paracrine_signaling” (adjusted *p* = 0.02210), “go_negative_regulation_of_epidermis_development” (adjusted *p* = 0.03244), and “go_negative_regulation_of_keratinocyte_differentiation” (adjusted *p* = 0.04440) gene sets in the recessive model (Appendix A). The 0–12 h plasma MEC was significantly associated with the “sotiriou_breast_cancer_grade_1_vs_3_dn” gene set in the additive model (adjusted *p* = 0.04805; Appendix A). The 0–6 h effect-site MEC was significantly associated with the “go_ccr2_chemokine_receptor_binding” and “go_paracrine_signaling” gene sets in the additive model (adjusted *p* = 0.01342 and 0.03030, respectively; Appendix A) and significantly associated with the “go_paracrine_signaling” and “go_ccr2_chemokine_receptor_binding” gene sets in the recessive model (adjusted *p* = 0.01030 and 0.02499, respectively; Appendix A). The 0–12 h effect-site MEC was significantly associated with the “pid_shp2_pathway” gene set in the recessive model (adjusted *p* = 0.02854; Appendix A). The genes that were included in these gene sets are listed in Appendix A. The *SERP2* gene, which was significantly associated with the phenotypes in the gene-based analysis, was not included in any of the gene sets (Appendix A). Among these genes, several genes were commonly included in two or three kinds of gene sets (Appendix A). Eight genes (*EZH2*, *GRHL2*, *HOXA7*, *MSX2*, *REG3A*, *REG3G*, *SRSF6*, and *TP63*) were commonly included in three kinds of gene sets (Appendix A).

## 3. Discussion

Although human genetic variants that are associated with human responsiveness and sensitivity to opioids have been explored by adopting opioid analgesics that are required for the treatment of disease-related pain, chronic pain, and perioperative/postoperative pain as the main endpoint [10,11,12,13,14,15,16,17,18,19,20,21,22,23,24,25,26,27,28,29,30,31,32,33], plasma and effect-site MECs have not been investigated in genetic studies, likely because of difficulties in measuring actual values of plasma and effect-site MECs. To comprehensively explore genetic factors that underlie large individual differences in fentanyl responsiveness and sensitivity after LAC, we first conducted a GWAS in this cohort of patients, focusing on plasma and effect-site MECs of fentanyl that were estimated with a pharmacokinetic simulation model [40,41,42]. As a result of the GWAS in surgical patients, 8–28 SNPs were selected as the top candidate SNPs that were significantly associated with a plasma or effect-site MEC that was averaged over the 0–6 h or 0–12 h postoperative period after LAC in all of the additive, dominant, and recessive models (Table 1 and Table 2; Appendix A). Among these, the rs966775 SNP that mapped to 5p13 had highly significant associations with 0–6 h plasma and 0–12 h effect-site MECs (Table 1 and Table 2). A gene that is located near the region of this SNP was *DRD1*, which encodes the dopamine D_1_ receptor. Altogether, our data indicated that the rs966775 SNP near the *DRD1* gene significantly affected fentanyl sensitivity. Compared with non-carriers, G-allele carriers of this SNP were associated with higher plasma and/or effect-site MECs of fentanyl, suggesting that G allele carriers would feel pain at a higher plasma/effect-site fentanyl concentration and thus would require more frequent self-dosing of fentanyl for adequate pain control. Although we acknowledge that the sample size of 350 patients may not be sufficiently large to draw definitive conclusions about genetic markers that contribute to individual differences in the MEC of fentanyl and that further research is needed with larger sample sizes and greater statistical power to validate our findings, the present results suggest that the rs966775 SNP could serve as a marker that predicts the efficacy of opioid analgesics for the treatment of postoperative pain.

In clinical postoperative pain management using patient-controlled analgesia (PCA), continuous pain relief should be achieved if the plasma opioid concentration is maintained in excess of the MEC, whereas pain will return if it decreases to the MEC. Thus, the MEC is indicated by the need for an additional intravenous (i.v.) opioid because of recurring pain [4,5]. The MEC of opioids varies depending on the type of surgery and intensity of postoperative pain. It gradually decreases with a decreasing intensity of postoperative pain [3,4,5,6]. Nevertheless, the MEC remains relatively constant within each patient over the postoperative period but varies widely among patients even after the same type of surgery [3,4,5]. Associations between genetic variants and MECs of opioids have not been investigated in genetic studies, likely because of difficulties in repeatedly measuring actual plasma opioid concentrations. However, opioids act on the effect site and not on plasma, and pharmacokinetic simulation models can predict plasma concentrations with acceptable accuracy [35,45,46] and estimate effect-site concentrations that are not measurable in humans [34,36,37]. Therefore, simulation models have been widely used in clinical studies [6,34,35,36,37,38,39], including one that evaluated the plasma MEC of fentanyl [6]. Using MECs of fentanyl that were determined with a simulation model, we conducted a GWAS and found that the rs966775 SNP significantly affected fentanyl sensitivity.

As mentioned above, we conducted GWASs of phenotypes that are related to opioid sensitivity and candidate gene studies [15,16,17,18,19,20,21,22,23,24,25,26,27,28,29,30,31,32,33]. Several candidate SNPs were found to be associated with phenotypes that are related to opioid sensitivity/pain. Among these are the rs2076222 SNP in the *LAMB3* gene region, which was associated with postoperative 24 h fentanyl requirements in subjects who underwent LAC. Although this SNP was also found to be nominally significantly associated with the estimated plasma and effect-site MECs in the present study in the additive model (combined *β* = −0.07784, nominal *p* = 0.00232, for the 0–6 h plasma MEC; combined *β* = −0.07517, nominal *p* = 0.00310, for the 0–12 h plasma MEC; combined *β* = −0.09432, nominal *p* = 0.00250, for the 0–6 h effect-site MEC; combined *β* = −0.08721, nominal *p* = 0.00369, for the 0–12 h effect-site MEC) and in the recessive model (combined *β* = −0.14420, nominal *p* = 0.00409, for the 0–6 h plasma MEC; combined *β* = −0.13780, nominal *p* = 0.00581, for the 0–12 h plasma MEC; combined *β* = −0.17340, nominal *p* = 0.00469, for the 0–6 h effect-site MEC; combined *β* = −0.15910, nominal *p* = 0.00706, for the 0–12 h effect-site MEC), the nominally significant associations would likely be attributable to the strong correlation among the values for postoperative 24 h fentanyl requirements and estimated plasma and effect-site MECs in the present study. The associations between other candidate SNPs that we identified in previous studies as candidates for opioid sensitivity and the estimated plasma and effect-site MECs in the present study were not even nominally significant (details not shown). These results might indicate the general difficulty in replicating results of human genetic association studies or reflect phenotypical differences between postoperative analgesic requirements per se and the plasma and effect-site MECs that were estimated with a pharmacokinetic simulation model in the present study.

The best candidate SNP in the present study was rs966775, which was found in the intergenic region. The protein-coding gene on chromosome 5 that was nearest to this SNP site was the *DRD1* gene. This gene encodes the dopamine D_1_ receptor, which is the most abundant dopamine receptor in the central nervous system. Although the D_1_ receptor has been shown to be involved in mechanisms of opioid analgesia in animal studies [47,48,49,50], the impact of this SNP on the expression and function of the *DRD1* gene product is not known but presumably may not be profound because this SNP is located more than 100 kbp from the gene region (Appendix A). The rs966775 SNP has not been previously reported to be associated with any phenotypes to date. Although this SNP was not in strong LD (*r*^2^ ≥ 0.80) with any other neighboring SNPs in our data (Appendix A), when these SNPs were referenced in HaploReg v. 4.1 and SNPinfo Web Server (accessed on 30 January 2023) [51,52], they were in strong LD with the rs7725278, rs897747, rs2382021, rs2890873, rs3955076, rs76895738, and rs10060502 SNPs (*r*^2^ ≥ 0.80) and were moderately linked to the rs12652255 SNP (*r*^2^ = 0.68) in Asian populations, including Japan. HaploReg v. 4.1 also showed that the rs966775 SNP could change six motifs for DNA-binding proteins and overlaps with an enhancer in the fat and skin. Nevertheless, none of these SNPs were significantly associated with mRNA expression levels of any genes in any tissues according to the GTEx portal (accessed on 30 January 2023) [53], suggesting that it is unlikely that these SNPs influence variations in opioid sensitivity among individuals by influencing the mRNA expression of some genes. The rs12652255 SNP was reported to be associated with the efficacy of Drotrecogin alfa, a drug with antithrombotic, profibrinolytic, anti-inflammatory, and cytoprotective properties in patients with severe sepsis [54], but its contribution to the efficacy of opioid analgesics remains unknown.

Among the candidate SNPs that were selected in our GWAS for the 0–12 h plasma MEC in the dominant model, the rs9533839 SNP was included, which was annotated as the *SERP2* gene (Appendix A). This gene was also significantly associated with the same trait (Appendix A) and the 0–6 h plasma and 0–12 h effect-site MECs (Table 3 and Table 4) in the gene-based analysis. The *SERP2* gene encodes stress-associated endoplasmic reticulum protein family member 2 (SERP2), which is predicted to be involved in the endoplasmic reticulum unfolded protein response and protein glycosylation. Although *SERP2* mRNA is known to be highly expressed in the brain, followed by the testis, according to the National Center for Biotechnology Information (NCBI) database, the functional relationship between this protein and the opioid system is unknown. In human cytogenetic studies, microdeletions of the *SERP2* gene were reported to be associated with acute lymphoblastic leukemias in children with Down syndrome [55], and focal deletions of this gene were also identified in 2–6% of adult cases of acute lymphoblastic leukemia [56]. However, no genetic variants, such as SNPs, in this gene region have been reported to be associated with diseases or other phenotypes. HaploReg v. 4.1 showed that the rs9533839 SNP could change six motifs for DNA-binding proteins and overlaps with an enhancer in nine tissues, and this SNP was found to be significantly associated with mRNA expression levels of the *TUSC8* gene in the prostate, breast (mammary tissue), and minor salivary gland according to the GTEx portal (accessed on 30 January 2023). The *TUSC8* gene encodes a non-coding RNA (ncRNA), TUSC8, and this ncRNA reportedly enhances the cisplatin sensitivity of non-small-cell lung cancer cells by regulating vascular endothelial growth factor A (VEGFA) [57], although the involvement of this ncRNA in opioid sensitivity remains unknown.

In the gene-set analysis, several significant associations were also found (Appendix A). Some of the genes that were included in the gene sets were included in two or three kinds of gene sets (Appendix A). Among the gene sets that were included in two kinds of gene sets, the *VEGFA* gene (Appendix A) is notable because VEGF-A protein, which is encoded by the *VEGFA* gene, is known to be involved in angiogenesis through activation of the opioid system [58], although opioids could also exert a proangiogenic effect at low doses but an antiangiogenic (toxic) effect at high doses [59]. Intriguingly, the *ANGPT1* gene was included in the “pid_shp2_pathway” gene set, which was significantly associated with the 0–12 h effect-site MEC in the recessive model (Appendix A). The *ANGPT1* gene encodes angiopoietin-1, a secreted glycoprotein that is a member of the angiopoietin family. Angiopoietin-1 is also known to be involved in angiogenesis. Mice that were engineered to lack angiopoietin-1 exhibited angiogenic deficits [60]. Although more studies are required, angiogenesis, with the involvement of angiopoietin-1, could also be modulated by actions of opioids, and the rs1283671 and rs1283720 SNPs within this gene region were found to be significantly associated with average daily opioid requirements for the treatment of cancer pain in our previous GWAS [33].

## 4. Materials and Methods

### 4.1. Patients

#### 4.1.1. Patients Who Underwent LAC

Enrolled in the study were 351 adult patients (20–85 years old, 218 males and 133 females) without severe coexisting systemic disease (American Society of Anesthesiologists Physical Status [ASA-PS] I or II) who were scheduled to undergo LAC for colon or rectal cancer at Saitama Medical University International Medical Center. Excluded were patients with severe coexisting disease (ASA-PS ≥ III), those taking pain medication for chronic pain, and those who were unlikely to be able to use a PCA pump (e.g., because of dementia). All of the individuals who were included in the study were of Japanese origin. Peripheral blood samples were collected from these subjects for gene analysis. Detailed demographic and clinical data of the subjects are provided in Appendix A and our previous reports [22,25].

The study was conducted according to guidelines of the Declaration of Helsinki and approved by the Institutional Review Board or Ethics Committee of Saitama Medical University International Medical Center and Tokyo Metropolitan Institute of Medical Science (Tokyo, Japan). Written informed consent was obtained from all of the patients.

#### 4.1.2. Surgical Protocol and Clinical Data

The protocols for anesthesia, surgery, and postoperative pain management and clinical data are detailed in our previous reports [22,25]. Briefly, general anesthesia was induced with fentanyl (0.1 mg), propofol (1–2 mg/kg), and rocuronium (0.8 mg/kg). After tracheal intubation, the inhalation of sevoflurane (1.5% in inspired concentration) and continuous infusion of remifentanil (0.25 µg/kg/min) were started. General anesthesia was thus maintained with sevoflurane, remifentanil, and rocuronium. At the end of surgery, remifentanil and sevoflurane were discontinued, and fentanyl (usually ≥ 0.1 mg) was given for immediate postoperative pain relief. The average remifentanil infusion rate (in µg/kg/min) during surgery was calculated by dividing the total dose of remifentanil that was required during surgery by the duration of surgery and body weight. When patients complained of even mild abdominal pain, fentanyl was given in increments of 0.05 mg until sufficient pain relief was achieved.

Postoperative pain was then managed with i.v. fentanyl PCA using a PCA pump (CADD-Legacy Model 6300, Smiths Medical Japan, Tokyo, Japan) that was filled with 1000 μg fentanyl diluted with normal saline to a total volume of 100 mL. The demand dose, dose lockout time, maximum allowable demand dose per hour, and continuous rate were set at 20 μg (2 mL), 5 min, 12 times (240 μg), and zero, respectively. Patient-controlled analgesia was principally continued for 24 h postoperatively. In cases of inadequate analgesia, i.v. flurbiprofen axetil (50 mg) or pentazocine (30 mg) was administered as a rescue analgesic. Severe postoperative nausea and vomiting were treated with i.v. droperidol (2.5 mg) or metoclopramide (10 mg).

Postoperative pain at rest was assessed on an 11-point numerical rating scale (0, no pain; 10, the worst pain imaginable). Sedation was assessed on a 4-point scale (0, awake and alert; 1, drowsy; 2, mostly asleep but easy to rouse; 3, asleep and difficult to rouse). Postoperative nausea and vomiting were assessed on a 4-point scale (0, no nausea or vomiting; 1, mild nausea; 2, severe nausea; 3, retching or vomiting). Postoperative pain scores, sedation scores, postoperative nausea and vomiting scores, respiratory rates, the cumulative number of PCA doses that were actually given to the patient, and the cumulative number of PCA doses that were attempted were recorded on a data collection sheet 2, 4, 6, 12, 18, and 24 h after surgery.

PCA fentanyl consumptions over 6 h, 12 h, and 24 h periods were calculated as cumulative doses of fentanyl that were actually given to patients via the PCA pump during the first 6 h, 12 h, and 24 h postoperative periods, respectively. The 6 h, 12 h, and 24 h total postoperative fentanyl requirements were calculated as sums of the i.v. fentanyl dose that was given around the end of surgery and 6 h, 12 h, and 24 h PCA fentanyl consumptions, respectively. Patient-controlled analgesia fentanyl consumptions and total postoperative fentanyl requirements were normalized to body weight. The 6 h, 12 h, and 24 h numbers of locked out doses were the differences between the cumulative number of doses attempted and doses that were given at 6, 12, and 24 h after surgery, respectively.

Plasma and effect-site concentrations of fentanyl over the 24 h postoperative period were estimated in each patient using BeConSim Monitoring (http://www.masuinet.com; accessed on 1 January 2020; Appendix A)—a pharmacokinetic simulation program that was developed by Masui (2010) [40] (Appendix A) based on Shafer’s three compartment model [61]—by inputting relevant clinical data, including age, sex, height, weight, the fentanyl dose that was given around the end of surgery, and subsequent PCA fentanyl consumption profiles over the 24 h postoperative period. A pair of plasma and effect-site MECs of fentanyl were indicated by plasma and effect-site fentanyl concentrations that were estimated immediately before each self-dosing of fentanyl. All pairs of plasma and effect-site MECs in each patient were averaged over the 6 h, 12 h, and 24 h postoperative periods to determine pairs of average plasma and effect-site MECs over these periods, which were expressed as the 0–6 h, 0–12 h, and 0–24 h plasma and effect-site MECs, respectively. Because many patients had completely consumed PCA fentanyl (1000 μg) by 24 h postoperatively and not by 12 h postoperatively, 0–6 h and 0–12 h plasma and effect-site MECs and not 0–24 h plasma and effect-site MECs were used for the main study endpoints. Detailed clinical data of the subjects are detailed in Appendix A.

### 4.2. Whole-Genome Genotyping, Quality Control, and Gene-Based and Gene-Set Analyses

#### 4.2.1. Whole-Genome Genotyping and Quality Control

For patients who underwent LAC, 10 mL of venous blood was sampled during anesthesia for the later preparation of genomic DNA specimens. After total genomic DNA was extracted from whole-blood samples using standard procedures and the concentration was adjusted to 100 ng/μL, whole-genome genotyping was performed using the Infinium Assay II with an iScan system (Illumina, San Diego, CA, USA) according to the manufacturer’s instructions. A total of 921,239 SNP markers survived the entire quality control filtration process and were used for the GWAS (detailed in our previous report) [22]. Two kinds of BeadChips were used for genotyping 256 and 95 samples, respectively: HumanOmniExpressExome-8 v. 1.0 (total markers: 951,117) and HumanOmniExpressExome-8 v. 1.1 (total markers: 958,178). Approximately 926,000 SNP markers were commonly included in all of the BeadChips. Quality control was properly performed the same way as in our previous report [22,25].

For phenotypes of the estimated 0–6 h plasma MEC and 0–12 h effect-site MEC, log QQ *p*-value plots as a result of the GWAS for the combined 351 samples were subsequently drawn to check the pattern of the generated *p*-value distribution, in which the observed *p* values against the values that were expected from the null hypothesis of a uniform distribution, calculated as −log10 (*p* value), were plotted for each model. All of the plots were mostly concordant with the expected line (y = x), especially over the range of 0 < −log10 (*p* value) < 4 for each model, indicating no apparent population stratification of the samples that were used in the study (Appendix A).

#### 4.2.2. Gene-Based and Gene-Set Analyses

Gene-based and gene-set approaches were adopted with Multi-marker Analysis of GenoMic Annotation (MAGMA) v. 1.06 [43], which is also available on the Functional Mapping and Annotation of Genome-Wide Association Studies (FUMA GWAS) v. 1.3.9 platform [44], to better understand genetic backgrounds and molecular mechanisms that underlie complex traits, such as opioid sensitivity in patients who underwent LAC. To examine the combined relationship between all genetic markers in the gene and the phenotype, genetic marker data were aggregated to the level of full genes in the gene-based analysis. Similarly, individual genes were compiled into groupings of genes with similar biological, functional, or other properties for the gene-set analysis. Gene-set analyses can thus shed light on the role that particular biological pathways or cellular processes may play in the genetic basis of a trait [43]. In these analyses, associations were explored for genes on autosomes 1–22 and the X chromosome, and the window of the genes to assign SNPs was set to 20 kb, thereby assigning SNPs within the 20 kb window of the gene (both sides) to that gene. For the reference panel, the 1000 Genome Phase3 EAS population was selected (http://ftp.1000genomes.ebi.ac.uk/vol1/ftp/release/20130502; accessed on 18 January 2023). In the gene-set analysis, gene sets were defined using the Molecular Signatures Database (MSigDB) v. 7.5.1 (https://www.gsea-msigdb.org/gsea/msigdb; accessed on 18 January 2023) [62]. A total of 10,678 gene sets (curated gene sets: 4761, Gene Ontology [GO] terms: 5917) from MsigDB were tested. In both analyses, Bonferroni correction for multiple testing was performed for all tested genes and gene sets. Adjusted values of *p* < 0.05 in the results were considered significant. The FUMA GWAS platform was also used for the visualization of QQ plots for the GWAS results and Manhattan plots for the gene-based analysis results.

### 4.3. Statistical Analysis

A three-stage GWAS was conducted for patients who underwent LAC to investigate the association between opioid sensitivity after surgery and the 921,239 SNPs that met the quality control criteria in a total of 351 subjects (117, 117, and 117 subjects for the first-, second-, and final-stage analyses, respectively) for whom postoperative clinical data were available, as described in our previous report [22]. As an index of opioid sensitivity after surgery, the estimated 0–6 h plasma MEC, 0–12 h plasma MEC, 0–6 h effect-site MEC, and 0–12 h effect-site MEC were used because these calculated values were expected to reflect the efficacy of fentanyl in each individual. Prior to the analyses, the quantitative values (ng/mL) were natural-log-transformed for approximation to the normal distribution according to the following formula: Value for analyses = Ln (1 + MEC value [ng/mL]). To explore the association between the SNPs and phenotypes, linear regression analyses were conducted in each stage of the analysis, in which the MEC value (ng/mL; log-transformed) and the genotype data for each SNP were incorporated as dependent and independent variables, respectively, with covariates that were found to be strongly associated with the dependent variable in a preliminary study. Male genotypes were not included in the analysis of X chromosome markers, whereas both male and female individuals were included in the association study for autosomal markers. Additive, dominant, and recessive genetic models for each minor allele were used for the analyses because of the previously insufficient knowledge about genetic factors that are associated with opioid sensitivity. The GWAS procedure is summarized in Appendix A for the 0–6 h plasma MEC and 0–12 h effect-site MEC, and the procedure was similar for the 0–12 h plasma MEC and 0–6 h effect-site MEC (details not shown). In the first-stage analysis of 117 subjects, the SNPs that showed statistical *p* < 0.05 were selected as candidate SNPs for the second-stage analysis among the 921,239 SNPs. For these SNPs, the second-stage analysis was conducted, and SNPs that showed *p* < 0.05 for the single analysis of this stage and combined analysis of the first and second stages were considered possible candidates. Similarly, the final-stage analysis was conducted by setting the threshold *p* values at 0.05, in which the SNPs that showed *p* < 0.05 for the single analysis of this stage and combined analysis of the first, second, and final stages were considered possible candidates. The potent SNPs were selected from these SNPs after LD-based SNP pruning to remove redundant SNPs due to strong LD (threshold *r*^2^ = 0.8) with each other, as conducted in a previous report [63]. In the final stage, *q* values of the false discovery rate (FDR) were also calculated to correct for multiple testing for the SNPs that were selected after the second-stage analysis and LD-based SNP pruning, based on previous reports [64,65]. The SNPs that showed *q* < 0.05 in the analysis among the SNPs that were selected after the final stage were considered to be genome-wide significant. Hardy–Weinberg equilibrium was additionally tested using Exact Tests for genotypic distributions of SNPs that were significantly associated with the phenotype. To calculate *q* values, Stratified False Discovery Rate (SFDR) software (http://www.utstat.toronto.edu/sun/Software/SFDR/index.html; accessed on 18 January 2023) was used [64,65,66]. All of the statistical analyses for genetics were performed using gPLINK v. 2.050, PLINK v. 1.07 (http://zzz.bwh.harvard.edu/plink/index.shtml; accessed on 18 January 2023) [67], and Haploview v. 4.2 (https://www.broadinstitute.org/haploview/haploview; accessed on 18 January 2023) [68]. Additionally, correlation analysis, the Mann–Whitney test, and linear regression analysis were performed for statistical analyses of clinical variables using SPSS Statistics v. 25 software (IBM, Armonk, NY, USA). For the statistical analyses of clinical variables, the criterion for significance was set at *p* < 0.05.

### 4.4. Additional in Silico Analysis

#### 4.4.1. Power Analysis

Statistical power analyses were preliminarily performed using G*Power v. 3.0.5 [69] as previously described [22,25]. Power analyses for the linear regression analyses revealed that the expected power (1 minus type II error probability) was 98.6% for Cohen’s conventional “medium” effect size of 0.15 [70] when the type I error probability was set at 0.05 and sample sizes were 117, corresponding to the sample size of each stage analysis in the present study. However, for the same type I error probability and sample sizes of 117, the expected power decreased to 32.9% when Cohen’s conventional “small” effect size was 0.02. Conversely, the estimated effect sizes were 0.0682 for the same type I error probability and sample sizes of 117 to achieve 80% power. Therefore, a single analysis in the present study was expected to detect true associations with the phenotype with 80% statistical power for effect sizes from large to moderately small, but not too small, although the exact effect size has been poorly understood in cases of SNPs that significantly contribute to opioid sensitivity.

#### 4.4.2. Linkage Disequilibrium Analysis

The LD analysis was performed using Haploview v. 4.2 [68] for a total of 351 samples from patients who underwent LAC for the genomic position from ~174,760,000 to ~174,900,000 on chromosome 5 (GRCh37) that includes both the rs966775 SNP and *DRD1* gene and its flanking region to identify relationships between SNPs in this region. The commonly used *D*′ and *r*^2^ values were pairwise calculated using the genotype dataset for each SNP to estimate the strength of LD between SNPs. Linkage disequilibrium blocks were defined as in a previous study [71]. For the visualization of LD plots with information about genomic position and related gene transcripts, the LDmatrix tool was also used (https://ldlink.nci.nih.gov/?tab=ldmatrix; accessed on 22 January 2023).

#### 4.4.3. Reference of Databases

Several databases and bioinformatic tools were referenced to more thoroughly examine the candidate SNP that may be related to human opioid analgesic sensitivity, including the NCBI database (http://www.ncbi.nlm.nih.gov; accessed on 19 January 2023), HaploReg v. 4.1 (https://pubs.broadinstitute.org/mammals/haploreg/haploreg.php; accessed on 19 January 2023) [51], SNPinfo Web Server (https://snpinfo.niehs.nih.gov; accessed on 19 January 2023) [52], and Genotype-Tissue Expression (GTEx) portal (https://gtexportal.org/home/; accessed on 19 January 2023) [53]. HaploReg is a tool for investigating non-coding genomic annotations at variations in haplotype blocks, such as potential regulatory SNPs at disease-associated sites [51]. The SNPinfo Web Server is a set of web-based SNP selection tools (freely available at https://snpinfo.niehs.nih.gov; accessed on 19 January 2023) where investigators can specify genes or linkage regions and select SNPs based on GWAS results, LD, and predicted functional characteristics of both coding and non-coding SNPs [52]. The GTEx project, an ongoing effort to create a comprehensive public resource to research tissue-specific gene expression and regulation [53], is the basis for the GTEx portal, which offers open access to such data as gene expression, quantitative trait loci, and histology images.

## 5. Conclusions

In conclusion, our GWASs revealed that the rs966775 SNP and *SERP2* gene were significantly associated with estimated plasma MECs over the 0–6 h and 0–12 h postoperative periods of fentanyl that was administered for the treatment of postoperative pain in LAC patients. Although the present results need to be corroborated by more research with larger sample sizes and greater statistical power, these findings indicate that the rs966775 SNP near the *DRD1* and *SERP2* genes could serve as markers that predict the efficacy of opioid analgesics for the treatment of postoperative pain.

## Figures and Tables

**Figure 1 ijms-24-08421-f001:**
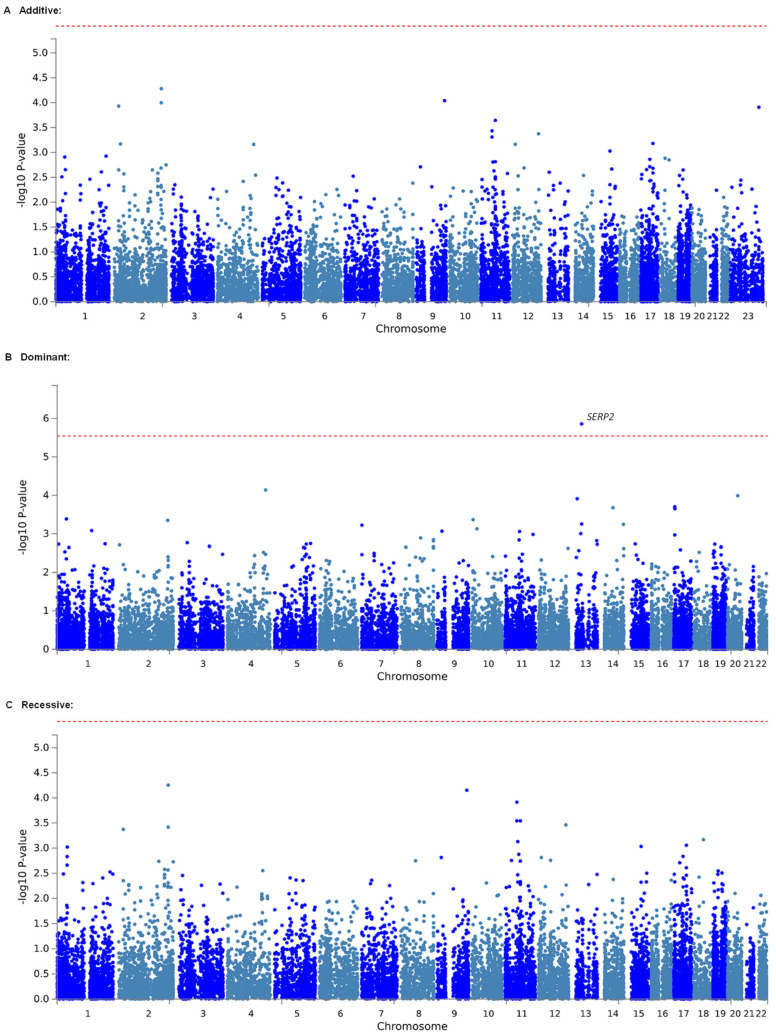
Manhattan plot of results of gene-based analysis for the 0–6 h plasma MEC. (**A**) Analytical plot in the additive model. (**B**) Analytical plot in the dominant model. (**C**) Analytical plot in the recessive model. The dashed red line shows the significant association threshold.

**Figure 2 ijms-24-08421-f002:**
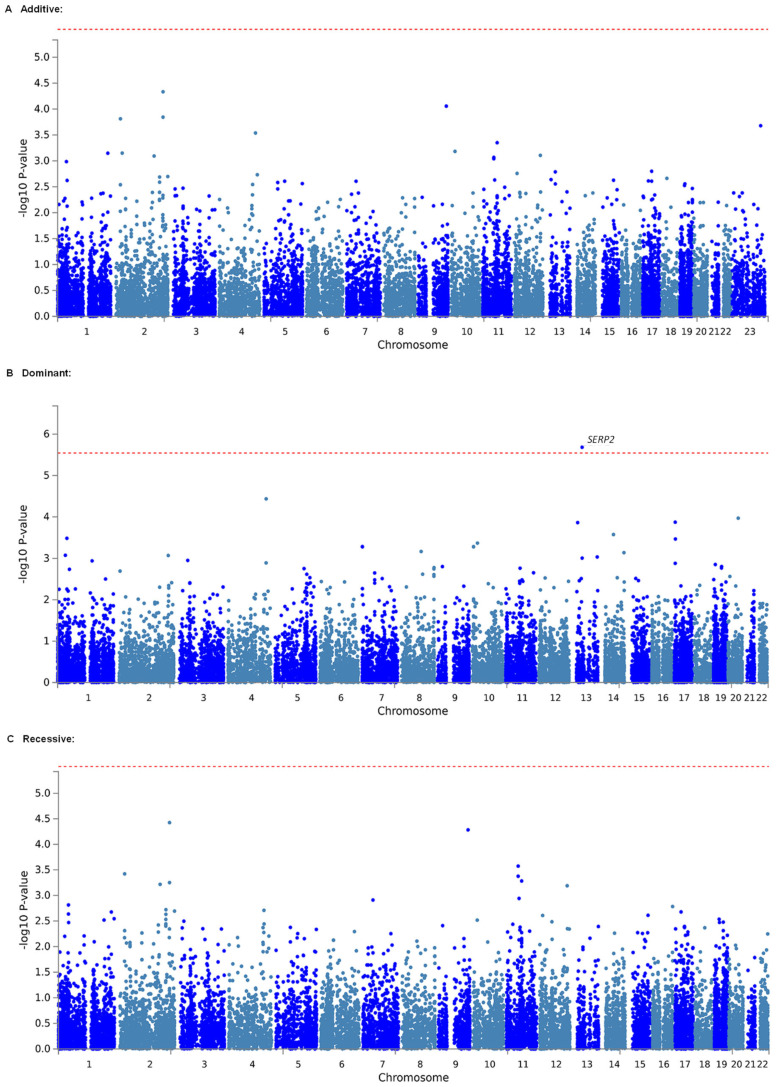
Manhattan plot of results of gene-based analysis for the 0–12 h effect-site MEC. (**A**) Analytical plot in the additive model. (**B**) Analytical plot in the dominant model. (**C**) Analytical plot in the recessive model. The dashed red line shows the significant association threshold.

**Table 1 ijms-24-08421-t001:** Top candidate SNPs selected from three-stage GWAS for the 0–6 h plasma MEC.

Model	Rank	SNP	CHR	Position		1st Stage			2nd Stage			Final Stage				Combined			Related Gene
						*β*	*p*		*β*	*p*		*β*	*p*	*q*		*β*	*p*		
Additive	1	rs966775	5	174,763,322		0.09569	0.004531		0.07693	0.03027		0.09776	0.0001217	0.0487 *		0.09157	0.0000001027		(*DRD1*)
Additive	2	rs6041532	20	12,652,435		0.2314	0.00159		0.1561	0.0334		0.2568	0.01035	0.4232		0.2003	0.000009788		-
Additive	3	rs9354118	6	95,147,902		0.05395	0.02689		0.06611	0.01174		0.06085	0.02056	0.5427		0.06091	0.00002527		-
Additive	4	rs9342409	6	95,098,682		0.05264	0.02903		0.06602	0.01574		0.05802	0.02816	-		0.06005	0.00004037		-
Additive	5	rs4806716	19	54,639,868		−0.05613	0.02562		−0.06506	0.02383		−0.06482	0.01373	0.4577		−0.06019	0.00006081		-
Additive	6	rs43211	19	54,652,203		−0.05488	0.02417		−0.08097	0.005927		−0.05655	0.03242	0.6175		−0.05957	0.00006086		*CNOT3*
Additive	7	rs9363197	6	95,104,559		0.05395	0.02689		0.05782	0.03145		0.05802	0.02816	-		0.05795	0.00006976		-
Additive	8	rs4764074	12	14,428,118		−0.08212	0.01157		−0.05914	0.04189		−0.05778	0.04175	0.7207		−0.06609	0.0000889		-
Additive	9	rs2676289	17	62,705,738		−0.1448	0.04901		−0.08609	0.03379		−0.1407	0.005126	0.4232		−0.1114	0.00008892		-
Additive	10	rs2759632	10	10,218,843		−0.09777	0.02416		−0.09776	0.04		−0.09442	0.01637	0.4926		−0.09548	0.00009947		-
Additive	11	rs1016214	20	16,992,615		0.1305	0.02942		0.2103	0.04618		0.1138	0.006869	0.4232		0.124	0.0001183		-
Additive	12	rs452325	8	88,505,366		0.06272	0.01896		0.06914	0.02194		0.05307	0.04622	-		0.05799	0.0001731		-
Additive	13	rs936229	15	75,132,319		0.07303	0.02838		0.06454	0.02857		0.06169	0.04701	0.7207		0.06311	0.0003549		*ULK3*
Additive	14	rs652930	1	201,628,577		0.091	0.03362		0.1072	0.02579		0.183	0.009896	0.4232		0.1015	0.0003897		*NAV1*
Dominant	1	rs6502266	17	13,395,720		0.1232	0.001448		0.1076	0.005443		0.08215	0.02588	0.4925		0.1004	0.000002452		-
Dominant	2	rs9889837	17	13,392,473		0.1232	0.001448		0.1076	0.005443		0.07843	0.03363	-		0.09947	0.000003073		-
Dominant	3	rs6481157	10	57,099,471		−0.1346	0.0009998		−0.0775	0.04471		−0.08874	0.01887	0.4925		−0.0947	0.00001424		-
Dominant	4	rs17738087	15	26,905,021		−0.1148	0.009143		−0.1091	0.01244		−0.09608	0.01112	0.4925		−0.1004	0.00002		*GABRB3*
Dominant	5	rs17081058	13	25,267,734		−0.09808	0.01557		−0.09338	0.02287		−0.08478	0.03985	0.4925		−0.0972	0.0000271		*ATP12A*
Dominant	6	rs1195916	12	131,503,109		0.08309	0.04349		0.07868	0.04677		0.1225	0.003758	0.4925		0.09485	0.00003543		*GPR133*
Dominant	7	rs172399	7	9,154,302		0.1075	0.007509		0.08153	0.0404		0.08112	0.0266	0.4925		0.08836	0.00005485		-
Dominant	8	rs13278423	8	87,720,419		−0.1184	0.00618		−0.09612	0.02496		−0.08617	0.03348	0.4925		−0.09586	0.00006912		*CNGB3*
Dominant	9	rs1160226	13	25,271,434		−0.08072	0.04288		−0.09013	0.02275		−0.07819	0.03853	0.4925		−0.08602	0.00009199		*ATP12A*
Dominant	10	rs3133206	18	57,237,478		0.08225	0.04547		0.09069	0.02442		0.09395	0.01793	0.4925		0.08851	0.00009607		*CCBE1*
Dominant	11	rs4963573	12	24,662,116		−0.09225	0.01659		−0.07917	0.03853		−0.0841	0.02294	0.4925		−0.0828	0.0001084		*SOX5*
Dominant	12	rs28350	3	42,418,446		0.1126	0.04845		0.129	0.02853		0.1208	0.01785	0.4925		0.1212	0.0001187		-
Dominant	13	rs10956972	8	87,768,331		0.1033	0.01486		0.08581	0.03306		0.1052	0.01004	-		0.08923	0.0001351		-
Dominant	14	rs1982563	8	87,776,019		0.1033	0.01486		0.08581	0.03306		0.1052	0.01004	0.4925		0.08923	0.0001351		-
Dominant	15	rs4940475	18	57,311,314		0.08404	0.04074		0.09424	0.02218		0.08181	0.03923	-		0.08696	0.0001466		*CCBE1*
Dominant	16	rs5766289	22	45,408,177		−0.08004	0.03874		−0.08407	0.02975		−0.0916	0.01366	0.4925		−0.08221	0.0001498		-
Dominant	17	rs1864309	18	57,309,059		0.08404	0.04074		0.09367	0.02112		0.08181	0.03923	-		0.08604	0.0001602		*CCBE1*
Dominant	18	rs1027804	8	18,919,857		−0.08584	0.03366		−0.07961	0.04826		−0.1072	0.005536	0.4925		−0.0851	0.0001687		-
Dominant	19	rs7592517	2	76,777,279		0.1278	0.002637		0.08475	0.03565		0.07594	0.04886	-		0.08143	0.0002555		-
Dominant	20	rs2139502	2	76,786,845		0.1278	0.002637		0.08475	0.03565		0.07594	0.04886	0.4925		0.08143	0.0002555		-
Dominant	21	exm−rs10873636	15	26,888,978		−0.09977	0.02877		−0.09658	0.03058		−0.07793	0.04325	-		−0.08463	0.0003904		*GABRB3*
Dominant	22	rs10873636	15	26,888,978		−0.09977	0.02877		−0.09658	0.03058		−0.07793	0.04325	-		−0.08463	0.0003904		*GABRB3*
Dominant	23	rs1863459	15	26,892,676		−0.09977	0.02877		−0.09658	0.03058		−0.07793	0.04325	0.4925		−0.08463	0.0003904		*GABRB3*
Dominant	24	rs6667463	1	175,518,442		−0.08513	0.03978		−0.115	0.01136		−0.08208	0.04367	0.4925		−0.0833	0.0004116		*TNR*
Dominant	25	rs12580224	12	71,086,426		0.1011	0.01189		0.08249	0.04176		0.07857	0.04728	0.4925		0.07887	0.0004391		*PTPRR*
Dominant	26	rs11945758	4	118,667,234		0.09923	0.0248		0.09422	0.02897		0.088	0.02815	0.4925		0.08344	0.0004876		-
Recessive	1	rs966775	5	174,763,322		0.1704	0.008618		0.1598	0.01971		0.1655	0.0005068	0.2027		0.1657	0.0000004313		(*DRD1*)
Recessive	2	rs6041532	20	12,652,435		0.4645	0.001396		0.3096	0.03374		0.5211	0.008986	0.3114		0.4019	0.000008521		-
Recessive	3	rs9354118	6	95,147,902		0.09027	0.03077		0.09682	0.03613		0.1358	0.001557	0.3116		0.1064	0.0000166		-
Recessive	4	rs9342409	6	95,098,682		0.08627	0.03752		0.09933	0.04201		0.1358	0.001557	0.3865		0.1067	0.00002053		-
Recessive	5	rs43211	19	54,652,203		−0.08789	0.04048		−0.1426	0.009216		−0.1305	0.005266	0.3116		−0.1106	0.00003108		*CNOT3*
Recessive	6	rs4764074	12	14,428,118		−0.149	0.01646		−0.1074	0.04912		−0.1277	0.01546	0.3116		−0.1258	0.00007724		-
Recessive	7	rs2759632	10	10,218,843		−0.1855	0.03017		−0.1941	0.03807		−0.2034	0.009063	0.693		−0.1905	0.00008301		-
Recessive	8	rs2146423	9	4,657,040		0.1379	0.01801		0.1222	0.03914		0.1779	0.004626	0.3116		0.1307	0.0001066		*C9orf68*
Recessive	9	rs12714409	2	596,532		0.09831	0.03269		0.1058	0.03244		0.1182	0.01311	0.3746		0.1038	0.0001265		-
Recessive	10	rs2642589	10	71,513,647		0.2137	0.0422		0.1943	0.03691		0.2436	0.03985	-		0.2132	0.000292		-

CHR, chromosome number; Position, chromosomal position (bp); *q*, *q* value for FDR correction of multiple comparison; Related gene, the nearest gene from the SNP site. * Significant after FDR correction (*q* < 0.05).

**Table 2 ijms-24-08421-t002:** Top candidate SNPs selected from three-stage GWAS for the 0–12 h effect-site MEC.

Model	Rank	SNP	CHR	Position		1st Stage			2nd Stage			Final Stage				Combined			Related Gene
						*β*	*p*		*β*	*p*		*β*	*p*	*q*		*β*	*p*		
Additive	1	rs966775	5	174,763,322		0.1089	0.004956		0.09541	0.02628		0.1153	0.0001176	0.0487 *		0.1071	0.0000001299		(*DRD1*)
Additive	2	rs6041532	20	12,652,435		0.2568	0.002112		0.1767	0.04689		0.2887	0.01417	0.5852		0.2257	0.00002233		-
Additive	3	rs9354118	6	95,147,902		0.05787	0.03746		0.07979	0.01181		0.07682	0.01282	0.5852		0.07173	0.00002448		-
Additive	4	rs9342409	6	95,098,682		0.05643	0.04045		0.08091	0.01476		0.07346	0.01801	-		0.0711	0.00003707		-
Additive	5	rs452325	8	88,505,366		0.08469	0.005179		0.07778	0.03335		0.06598	0.03512	-		0.07321	0.00005313		-
Additive	6	rs391916	8	88,512,286		0.08987	0.003892		0.07335	0.04768		0.07024	0.03155	-		0.07531	0.00005547		-
Additive	7	rs9363197	6	95,104,559		0.05787	0.03746		0.07095	0.02872		0.07346	0.01801	-		0.06866	0.00006093		-
Additive	8	rs4764074	12	14,428,118		−0.08937	0.01592		−0.07677	0.02874		−0.06911	0.0377	0.62		−0.07886	0.00006821		-
Additive	9	rs4806716	19	54,639,868		−0.06474	0.02339		−0.07928	0.02267		−0.07113	0.02095	0.5852		−0.06999	0.00007154		-
Additive	10	rs375481	8	88,490,938		0.0844	0.005855		0.08185	0.02178		0.06467	0.03894	0.62		0.07169	0.00007405		-
Additive	11	rs463809	8	88,513,842		0.0884	0.004368		0.07265	0.04837		0.06353	0.04896	-		0.07236	0.00009161		-
Additive	12	rs12035559	1	34,499,921		−0.09324	0.04609		−0.08629	0.01417		−0.0688	0.03792	0.62		−0.08029	0.0001101		*CSMD2*
Additive	13	rs2759632	10	10,218,843		−0.1166	0.01793		−0.1207	0.03614		−0.09306	0.04611	0.6725		−0.1076	0.0001932		-
Additive	14	rs4759709	12	131,001,468		0.07097	0.04568		0.08022	0.02878		0.08548	0.0188	0.5852		0.07568	0.000232		*RIMBP2*
Additive	15	rs2013536	8	87,669,792		0.06629	0.03569		0.07622	0.02542		0.05973	0.04567	0.6725		0.06185	0.0005526		*CNGB3*
Additive	16	rs2160974	12	108,883,621		0.06169	0.04197		0.07592	0.02893		0.06844	0.03283	0.62		0.06312	0.0005903		-
Additive	17	rs2368473	17	32,534,215		0.172	0.04925		0.09229	0.04477		0.1336	0.04801	0.6725		0.09914	0.003209		-
Dominant	1	rs6502266	17	13,395,720		0.1431	0.001155		0.1187	0.01152		0.08856	0.04179	-		0.1128	0.000006864		-
Dominant	2	rs17081058	13	25,267,734		−0.1161	0.01188		−0.1097	0.02728		−0.1085	0.02525	0.7497		−0.1163	0.00001918		*ATP12A*
Dominant	3	rs10836454	11	4,696,875		−0.1235	0.03386		−0.1135	0.03507		−0.1552	0.002679	0.7497		−0.1302	0.00002082		-
Dominant	4	rs17738087	15	26,905,021		−0.1233	0.01412		−0.1238	0.01927		−0.1158	0.009327	0.7497		−0.1162	0.00002685		*GABRB3*
Dominant	5	rs751687	8	15,608,896		0.1227	0.004706		0.1077	0.02049		0.09564	0.02716	0.7497		0.1054	0.00002937		*TUSC3*
Dominant	6	rs9968875	6	151,313,367		0.2127	0.03445		0.2375	0.002204		0.2158	0.04357	0.7497		0.2128	0.00003786		*MTHFD1L*
Dominant	7	rs4131101	5	119,195,837		0.1201	0.01361		0.1293	0.01648		0.1006	0.03878	0.7497		0.1165	0.00004616		-
Dominant	8	rs746427	20	48,939,076		−0.1004	0.02234		−0.106	0.04067		−0.1111	0.01493	-		−0.1057	0.00004884		-
Dominant	9	rs4580854	6	15,025,298		−0.09948	0.02485		−0.1086	0.0188		−0.1015	0.0201	0.7497		−0.1021	0.00005343		-
Dominant	10	rs1431210	6	103,229,346		0.1369	0.01037		0.1107	0.02506		0.103	0.02836	-		0.1129	0.00005784		-
Dominant	11	rs2143500	20	45,253,237		0.08748	0.04958		0.1251	0.009653		0.103	0.02006	0.7497		0.1019	0.00008058		*SLC13A3*
Dominant	12	exm2270377	6	103,225,137		0.1369	0.01037		0.103	0.03793		0.103	0.02836	-		0.1101	0.00008819		-
Dominant	13	rs6020445	20	48,939,863		−0.1004	0.02234		−0.1111	0.03323		−0.09611	0.03785	0.7497		−0.102	0.00009374		-
Dominant	14	rs3935993	5	119,196,820		0.1066	0.02517		0.1293	0.01648		0.1006	0.03878	-		0.1107	0.00009776		-
Dominant	15	rs13195313	6	103,175,290		0.1276	0.0143		0.103	0.03793		0.1063	0.02584	-		0.1094	0.00009796		-
Dominant	16	rs12817917	12	5,321,651		0.1188	0.02369		0.1311	0.02116		0.116	0.04445	0.7497		0.1212	0.0001064		-
Dominant	17	rs1160226	13	25,271,434		−0.09663	0.03322		−0.1051	0.02826		−0.09048	0.04211	0.7497		−0.09989	0.0001114		*ATP12A*
Dominant	18	rs13278423	8	87,720,419		−0.1293	0.008762		−0.1109	0.03276		−0.1019	0.03276	0.7497		−0.1093	0.0001131		*CNGB3*
Dominant	19	rs7592517	2	76,777,279		0.1386	0.004283		0.1108	0.02292		0.1029	0.02308	-		0.1004	0.0001222		-
Dominant	19	rs2139502	2	76,786,845		0.1386	0.004283		0.1108	0.02292		0.1029	0.02308	0.7497		0.1004	0.0001222		-
Dominant	21	rs10956972	8	87,768,331		0.1137	0.01871		0.1022	0.03598		0.1251	0.009372	-		0.1041	0.0001519		-
Dominant	21	rs1982563	8	87,776,019		0.1137	0.01871		0.1022	0.03598		0.1251	0.009372	0.7497		0.1041	0.0001519		-
Dominant	23	rs4963573	12	24,662,116		−0.0974	0.02673		−0.09413	0.04208		−0.09436	0.03043	0.7497		−0.09361	0.0001979		*SOX5*
Dominant	24	rs12580224	12	71,086,426		0.1173	0.01037		0.1015	0.0382		0.0941	0.0436	0.7497		0.09522	0.0003004		*PTPRR*
Dominant	25	rs6792514	3	42,429,817		0.1291	0.04677		0.156	0.03299		0.1051	0.04156	0.7497		0.1183	0.0006295		-
Recessive	1	rs966775	5	174,763,322		0.1908	0.01057		0.1977	0.01704		0.1931	0.0005773	0.1068		0.1919	0.0000006958		(*DRD1*)
Recessive	2	rs6041532	20	12,652,435		0.5168	0.001824		0.3513	0.04671		0.5921	0.0118	0.3928		0.455	0.00001802		-
Recessive	3	rs43211	19	54,652,203		−0.1046	0.03218		−0.1699	0.01039		−0.1444	0.008904	0.3928		−0.1278	0.00004229		*CNOT3*
Recessive	4	rs4764074	12	14,428,118		−0.154	0.02998		−0.1442	0.02867		−0.1595	0.01012	0.3928		−0.1515	0.00005068		-
Recessive	5	rs2146423	9	4,657,040		0.1363	0.04086		0.1478	0.0392		0.2007	0.006745	0.3928		0.1479	0.0001913		*C9orf68*
Recessive	6	rs12714409	2	596,532		0.1057	0.04401		0.1324	0.02693		0.1247	0.02672	0.5038		0.1185	0.0001978		-
Recessive	7	rs2759632	10	10,218,843		−0.2173	0.02572		−0.237	0.03636		−0.2005	0.02973	0.5089		−0.2112	0.0002074		-
Recessive	8	rs2199503	3	119,778,489		0.1374	0.04339		0.1528	0.02812		0.1276	0.04407	0.5889		0.1352	0.0003717		*GSK3B*
Recessive	9	rs10486791	7	16,284,326		0.1441	0.04552		0.1725	0.04761		0.2051	0.02432	0.5038		0.1639	0.0003742		*ISPD, LOC100506025*
Recessive	10	rs2368473	17	32,534,215		0.3506	0.04403		0.1788	0.04874		0.2906	0.03274	0.5267		0.2039	0.002313		-

CHR, chromosome number; Position, chromosomal position (bp); *q*, *q* value for FDR correction of multiple comparison; Related gene, the nearest gene from the SNP site. * Significant after FDR correction (*q* < 0.05).

**Table 3 ijms-24-08421-t003:** Top 20 candidate genes selected from gene-based analysis for the 0–6 h plasma MEC.

Model	Rank	CHR	Gene Start Position	Gene Stop Position	Gene	nSNPs	Z Statistic	*p*	*p* ^a^
Additive	1	2	220,378,892	220,403,494	*ASIC4*	6	3.8796	0.00005232	0.90440352
Additive	2	9	132,500,610	132,515,326	*PTGES*	6	3.7421	0.000091253	1
Additive	3	2	220,299,568	220,363,009	*SPEG*	15	3.7182	0.00010034	1
Additive	4	2	20,448,452	20,551,995	*PUM2*	6	3.678	0.00011753	1
Additive	5	X	135,295,381	135,338,641	*MAP7D3*	6	3.6653	0.00012353	1
Additive	6	11	67,195,931	67,202,872	*RPS6KB2*	2	3.5064	0.00022708	1
Additive	7	11	51,515,282	51,516,211	*OR4C46*	1	3.3757	0.0003682	1
Additive	8	12	122,089,024	122,110,537	*MORN3*	3	3.3375	0.00042262	1
Additive	9	11	51,411,378	51,412,448	*OR4A5*	2	3.296	0.00049041	1
Additive	10	17	56,597,611	56,618,179	*45173*	5	3.2112	0.00066098	1
Additive	11	2	28,680,012	28,866,654	*PLB1*	66	3.2041	0.00067735	1
Additive	12	12	12,813,825	12,849,141	*GPR19*	11	3.1993	0.00068891	1
Additive	13	4	169,418,217	169,849,608	*PALLD*	119	3.1987	0.00069036	1
Additive	14	15	66,679,155	66,784,650	*MAP2K1*	6	3.1084	0.00094037	1
Additive	15	1	235,490,665	235,507,847	*GGPS1*	4	3.0385	0.0011887	1
Additive	16	1	41,157,320	41,237,275	*NFYC*	5	3.0258	0.0012398	1
Additive	17	18	24,432,002	24,445,782	*AQP4*	2	3.0094	0.001309	1
Additive	18	17	42,325,753	42,345,509	*SLC4A1*	10	2.9943	0.0013754	1
Additive	19	18	43,405,477	43,424,045	*SIGLEC15*	6	2.9848	0.0014187	1
Additive	20	11	67,202,981	67,205,538	*PTPRCAP*	1	2.9592	0.001542	1
Dominant	1	13	44,947,801	44,971,850	*SERP2*	5	4.6871	0.0000013857	0.02424975 *
Dominant	2	4	169,418,217	169,849,608	*PALLD*	134	3.7987	0.000072725	1
Dominant	3	20	45,186,463	45,304,714	*SLC13A3*	79	3.7129	0.00010247	1
Dominant	4	13	25,254,549	25,285,921	*ATP12A*	14	3.6666	0.00012288	1
Dominant	5	17	4,574,679	4,607,632	*PELP1*	4	3.5425	0.0001982	1
Dominant	6	14	57,936,019	57,960,585	*C14orf105*	7	3.5263	0.00021073	1
Dominant	7	17	5,402,747	5,522,744	*NLRP1*	37	3.5107	0.00022346	1
Dominant	8	1	41,157,320	41,237,275	*NFYC*	5	3.3447	0.00041192	1
Dominant	9	10	5,435,061	5,446,793	*TUBAL3*	7	3.3324	0.00043043	1
Dominant	10	2	218,148,742	218,621,316	*DIRC3*	104	3.3216	0.00044747	1
Dominant	11	13	45,007,655	45,151,283	*TSC22D1*	13	3.2597	0.00055764	1
Dominant	12	14	104,552,016	104,579,098	*ASPG*	6	3.254	0.00056887	1
Dominant	13	7	1,509,913	1,545,489	*INTS1*	5	3.2398	0.00059799	1
Dominant	14	10	22,823,778	23,003,484	*PIP4K2A*	47	3.1764	0.00074551	1
Dominant	15	1	153,389,000	153,395,701	*S100A7A*	1	3.1454	0.0008292	1
Dominant	16	9	22,002,902	22,009,362	*CDKN2B*	3	3.1361	0.00085605	1
Dominant	17	11	63,580,860	63,595,190	*C11orf84*	5	3.1307	0.0008719	1
Dominant	18	13	41,129,804	41,240,734	*FOXO1*	16	3.0931	0.00099042	1
Dominant	19	11	123,676,043	123,677,095	*OR6M1*	3	3.0782	0.0010414	1
Dominant	20	17	4,613,784	4,624,794	*ARRB2*	1	3.0692	0.001073	1
Recessive	1	2	220,378,892	220,403,494	*ASIC4*	6	3.8643	0.000055694	0.930702434
Recessive	2	9	132,500,610	132,515,326	*PTGES*	6	3.8064	0.000070511	1
Recessive	3	11	51,515,282	51,516,211	*OR4C46*	1	3.6687	0.0001219	1
Recessive	4	11	67,195,931	67,202,872	*RPS6KB2*	2	3.4441	0.00028648	1
Recessive	5	11	51,411,378	51,412,448	*OR4A5*	2	3.4435	0.00028713	1
Recessive	6	12	122,089,024	122,110,537	*MORN3*	3	3.3926	0.00034621	1
Recessive	7	2	220,299,568	220,363,009	*SPEG*	15	3.3645	0.0003834	1
Recessive	8	2	20,448,452	20,551,995	*PUM2*	6	3.3366	0.00042402	1
Recessive	9	18	43,405,477	43,424,045	*SIGLEC15*	6	3.202	0.00068241	1
Recessive	10	11	55,563,032	55,563,976	*OR5D14*	2	3.177	0.00074398	1
Recessive	11	17	56,597,611	56,618,179	*44808*	5	3.1273	0.00088212	1
Recessive	12	15	66,679,155	66,784,650	*MAP2K1*	6	3.1118	0.00092985	1
Recessive	13	1	45,240,923	45,244,451	*RPS8*	1	3.1037	0.0009556	1
Recessive	14	11	60,197,062	60,222,687	*MS4A5*	8	3.004	0.0013323	1
Recessive	15	17	41,717,756	41,739,322	*MEOX1*	5	2.9757	0.0014618	1
Recessive	16	1	44,398,992	44,402,913	*ARTN*	2	2.9727	0.0014762	1
Recessive	17	9	19,408,925	19,452,018	*ACER2*	9	2.9601	0.0015377	1
Recessive	18	12	12,813,825	12,849,141	*GPR19*	11	2.9597	0.0015395	1
Recessive	19	12	54,104,903	54,121,529	*CALCOCO1*	9	2.9198	0.0017513	1
Recessive	20	11	27,676,440	27,743,605	*BDNF*	10	2.9176	0.0017634	1

Model, the genetic model in which candidate genes were selected by analysis; CHR, chromosome number; nSNPs, number of SNPs annotated to the gene; Z Statistic, gene-based test statistic; *p*^a^, adjusted *p* value for multiple testing. * Significant association after Bonferroni correction.

**Table 4 ijms-24-08421-t004:** Top 20 candidate genes selected from gene-based analysis for the 0–12 h effect-site MEC.

Model	Rank	CHR	Gene Start Position	Gene Stop Position	Gene	nSNPs	Z Statistic	*p*	*p* ^a^
Additive	1	2	220,378,892	220,403,494	*ASIC4*	6	3.9085	0.000046429	0.802571694
Additive	2	9	132,500,610	132,515,326	*PTGES*	6	3.7514	0.000087928	1
Additive	3	2	220,299,568	220,363,009	*SPEG*	15	3.6267	0.00014354	1
Additive	4	2	20,448,452	20,551,995	*PUM2*	6	3.608	0.00015431	1
Additive	5	X	135,295,381	135,338,641	*MAP7D3*	6	3.5274	0.00020985	1
Additive	6	4	169,418,217	169,849,608	*PALLD*	119	3.4411	0.00028971	1
Additive	7	11	67,195,931	67,202,872	*RPS6KB2*	2	3.3222	0.0004466	1
Additive	8	10	18,240,768	18,332,221	*SLC39A12*	37	3.2132	0.00065641	1
Additive	9	2	28,680,012	28,866,654	*PLB1*	66	3.1911	0.00070872	1
Additive	10	1	235,490,665	235,507,847	*GGPS1*	4	3.1896	0.00071235	1
Additive	11	12	122,089,024	122,110,537	*MORN3*	3	3.162	0.00078353	1
Additive	12	2	178,477,720	178,483,694	*TTC30A*	6	3.1536	0.00080626	1
Additive	13	11	51,515,282	51,516,211	*OR4C46*	1	3.1356	0.0008575	1
Additive	14	11	51,411,378	51,412,448	*OR4A5*	2	3.118	0.00091033	1
Additive	15	1	41,157,320	41,237,275	*NFYC*	5	3.0802	0.0010343	1
Additive	16	17	42,325,753	42,345,509	*SLC4A1*	10	2.9505	0.0015861	1
Additive	17	13	44,947,801	44,971,850	*SERP2*	3	2.942	0.0016303	1
Additive	18	12	12,813,825	12,849,141	*GPR19*	11	2.9204	0.0017477	1
Additive	19	4	178,163,693	178,169,927	*RP11-487E13.1*	3	2.9023	0.0018523	1
Additive	20	2	242,673,994	242,708,231	*D2HGDH*	2	2.8772	0.0020061	1
Dominant	1	13	44,947,801	44,971,850	*SERP2*	5	4.6035	0.0000020769	0.03634575 *
Dominant	2	4	169,418,217	169,849,608	*PALLD*	134	3.9649	0.000036709	0.6424075
Dominant	3	20	45,186,463	45,304,714	*SLC13A3*	79	3.7005	0.00010758	1
Dominant	4	17	4,574,679	4,607,632	*PELP1*	4	3.6432	0.00013461	1
Dominant	5	13	25,254,549	25,285,921	*ATP12A*	14	3.6374	0.0001377	1
Dominant	6	14	57,936,019	57,960,585	*C14orf105*	7	3.4622	0.00026792	1
Dominant	7	1	41,157,320	41,237,275	*NFYC*	5	3.4055	0.00033018	1
Dominant	8	17	5,402,747	5,522,744	*NLRP1*	37	3.3951	0.00034301	1
Dominant	9	10	22,823,778	23,003,484	*PIP4K2A*	47	3.3317	0.00043156	1
Dominant	10	7	1,509,913	1,545,489	*INTS1*	5	3.2775	0.00052371	1
Dominant	11	10	5,435,061	5,446,793	*TUBAL3*	7	3.2772	0.00052424	1
Dominant	12	8	87,878,670	88,627,447	*CNBD1*	106	3.2034	0.00067909	1
Dominant	13	14	104,552,016	104,579,098	*ASPG*	6	3.1844	0.00072522	1
Dominant	14	1	33,979,609	34,631,443	*CSMD2*	246	3.1422	0.00083837	1
Dominant	15	2	218,148,742	218,621,316	*DIRC3*	104	3.1384	0.00084928	1
Dominant	16	13	112,240,548	112,324,955	*RP11-65D24.2*	22	3.1144	0.00092144	1
Dominant	17	13	45,007,655	45,151,283	*TSC22D1*	13	3.096	0.00098065	1
Dominant	18	3	38,029,550	38,048,679	*VILL*	6	3.0576	0.0011155	1
Dominant	19	1	153,389,000	153,395,701	*S100A7A*	1	3.0495	0.001146	1
Dominant	20	4	169,277,886	169,458,937	*DDX60L*	75	3.0157	0.0012818	1
Recessive	1	2	220,378,892	220,403,494	*ASIC4*	6	3.9596	0.000037539	0.627314229
Recessive	2	9	132,500,610	132,515,326	*PTGES*	6	3.8807	0.000052076	0.870242036
Recessive	3	11	51,515,282	51,516,211	*OR4C46*	1	3.4626	0.0002675	1
Recessive	4	2	20,448,452	20,551,995	*PUM2*	6	3.3671	0.00037976	1
Recessive	5	11	51,411,378	51,412,448	*OR4A5*	2	3.3374	0.00042284	1
Recessive	6	11	67,195,931	67,202,872	*RPS6KB2*	2	3.2775	0.00052365	1
Recessive	7	2	220,299,568	220,363,009	*SPEG*	15	3.2573	0.00056237	1
Recessive	8	2	178,477,720	178,483,694	*TTC30A*	6	3.235	0.00060813	1
Recessive	9	12	122,089,024	122,110,537	*MORN3*	3	3.2166	0.00064853	1
Recessive	10	11	55,563,032	55,563,976	*OR5D14*	2	3.0495	0.0011462	1
Recessive	11	7	44,836,279	44,864,163	*PPIA*	3	3.0277	0.0012321	1
Recessive	12	1	44,398,992	44,402,913	*ARTN*	2	2.9601	0.0015375	1
Recessive	13	16	88,519,725	88,603,424	*ZFPM1*	10	2.9384	0.0016497	1
Recessive	14	2	204,259,068	204,400,133	*RAPH1*	9	2.894	0.001902	1
Recessive	15	4	156,129,781	156,138,230	*NPY2R*	2	2.8848	0.0019584	1
Recessive	16	2	242,673,994	242,708,231	*D2HGDH*	2	2.874	0.0020268	1
Recessive	17	17	26,975,374	26,989,207	*SDF2*	1	2.8623	0.002103	1
Recessive	18	1	235,490,665	235,507,847	*GGPS1*	4	2.8614	0.0021089	1
Recessive	19	1	44,440,159	44,443,967	*ATP6V0B*	3	2.8318	0.002314	1
Recessive	20	2	204,192,942	204,312,446	*ABI2*	6	2.8279	0.0023427	1

Model, the genetic model in which candidate genes were selected by analysis; CHR, chromosome number; nSNPs, number of SNPs annotated to the gene; Z Statistic, gene-based test statistic; *p*^a^, adjusted *p* value for multiple testing. * Significant association after Bonferroni correction.

## Data Availability

Data that are presented in this study are available upon request from the corresponding author.

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
