# Peer review of "Genome-Wide Association Study Identifies Genetic Polymorphisms Associated with Estimated Minimum Effective Concentration of Fentanyl in Patients Undergoing Laparoscopic-Assisted Colectomy"

_ijms, 2023, doi:10.3390/ijms24098421_

Round 1

Reviewer 1 Report

The paper describes a three-stage genome-wide association study (GWAS) to identify potential genetic markers that contribute to individual differences in the minimum effective concentration (MEC) of fentanyl, an opioid used for postoperative pain management. The study involved 350 patients who underwent laparoscopic-assisted colectomy, and their plasma and effect site concentrations of fentanyl were estimated using a pharmacokinetic simulation model. The results of the GWAS showed that an intergenic SNP, rs966775, located near the DRD1 and SERP2 genes, had significant associations with the plasma and effect site MECs of fentanyl. The minor G allele of rs966775 was found to be associated with increased MECs of fentanyl. The gene-based analysis further showed that the SERP2 gene was significantly associated with the MECs of fentanyl.

While the paper presents an interesting study on the genetic basis of individual variations in the response to fentanyl, there are some aspects of the research that could be criticized:

1. the sample size of 350 patients may not be large enough to draw definitive conclusions about the genetic markers that contribute to individual differences in the MEC of fentanyl. A larger sample size could increase the statistical power of the study and help to validate the findings.

2.  the study only focused on patients who underwent laparoscopic-assisted colectomy, and it is unclear whether the findings can be generalized to other patient populations or surgical procedures. Further studies involving a wider range of patients and procedures are needed to determine the broader applicability of the findings.

Overall, the paper presents an interesting study that sheds light on the genetic basis of individual variations in the response to fentanyl. The use of a pharmacokinetic simulation model to estimate the MECs of fentanyl provides a robust and quantitative approach to studying this complex trait. The identification of rs966775 and the SERP2 gene as potential markers for predicting the efficacy of fentanyl in postoperative pain management has important clinical implications for personalized pain treatment. However, the authors acknowledge the need for further research with larger sample sizes to validate their findings.

The quality of English language in the paper is generally good. The sentences are well-structured and easy to understand, and the paper effectively communicates the key findings of the research. There are a few minor errors in grammar and punctuation, but these do not detract from the overall clarity of the paper. However, there are a few areas where the wording could be improved for greater precision or clarity. For example, in the conclusion, the phrase "the estimated plasma 0-6 h plasma MEC and 0-12 h effect site MEC" could be rephrased as "the estimated plasma MECs over the 0-6 hour and 0-12 hour postoperative periods". Overall, the paper is well-written and effectively communicates the research findings.

Author Response

Response to Reviewer 1

Comments and Suggestions for Authors

The paper describes a three-stage genome-wide association study (GWAS) to identify potential genetic markers that contribute to individual differences in the minimum effective concentration (MEC) of fentanyl, an opioid used for postoperative pain management. The study involved 350 patients who underwent laparoscopic-assisted colectomy, and their plasma and effect site concentrations of fentanyl were estimated using a pharmacokinetic simulation model. The results of the GWAS showed that an intergenic SNP, rs966775, located near the DRD1 and SERP2 genes, had significant associations with the plasma and effect site MECs of fentanyl. The minor G allele of rs966775 was found to be associated with increased MECs of fentanyl. The gene-based analysis further showed that the SERP2 gene was significantly associated with the MECs of fentanyl.

While the paper presents an interesting study on the genetic basis of individual variations in the response to fentanyl, there are some aspects of the research that could be criticized:

  1. the sample size of 350 patients may not be large enough to draw definitive conclusions about the genetic markers that contribute to individual differences in the MEC of fentanyl. A larger sample size could increase the statistical power of the study and help to validate the findings.

Response: As mentioned by the reviewer, the sample size of 350 patients may not be sufficiently large to draw definitive conclusions about genetic markers that contribute to individual differences in the MEC of fentanyl. Larger sample sizes would increase statistical power and help validate the findings. According to the reviewer’s suggestion, we mentioned this in the Discussion and Conclusions sections. The sample size of 350 patients for the analysis could be higher but was not increased in the present study because the present study was conducted as a reanalysis of samples that were analyzed in previous studies [22,25]. As a result, SNPs and genes that are significantly associated with phenotypes that are related to opioid sensitivity were identified in the present study, although not all of our previous GWASs identified significant associations [30]. Furthermore, the present study was conducted in patients who underwent only LAC. Postoperative opioid requirements may indeed be different among different types of surgery. As stated by the reviewer, future studies of subjects who undergo LAC or other surgeries with larger sample sizes are required to further substantiate the present findings. [Lines 291-296, 618]

  1. the study only focused on patients who underwent laparoscopic-assisted colectomy, and it is unclear whether the findings can be generalized to other patient populations or surgical procedures. Further studies involving a wider range of patients and procedures are needed to determine the broader applicability of the findings.

Overall, the paper presents an interesting study that sheds light on the genetic basis of individual variations in the response to fentanyl. The use of a pharmacokinetic simulation model to estimate the MECs of fentanyl provides a robust and quantitative approach to studying this complex trait. The identification of rs966775 and the SERP2 gene as potential markers for predicting the efficacy of fentanyl in postoperative pain management has important clinical implications for personalized pain treatment. However, the authors acknowledge the need for further research with larger sample sizes to validate their findings.

Response: According to the reviewer’s suggestion, we acknowledged the need for further research with larger sample sizes to validate our findings in the Discussion section. [Lines 291-296]

Comments on the Quality of English Language

The quality of English language in the paper is generally good. The sentences are well-structured and easy to understand, and the paper effectively communicates the key findings of the research. There are a few minor errors in grammar and punctuation, but these do not detract from the overall clarity of the paper. However, there are a few areas where the wording could be improved for greater precision or clarity. For example, in the conclusion, the phrase "the estimated plasma 0-6 h plasma MEC and 0-12 h effect site MEC" could be rephrased as "the estimated plasma MECs over the 0-6 hour and 0-12 hour postoperative periods". Overall, the paper is well-written and effectively communicates the research findings.

Response: According to the reviewer’s suggestion, we rephrased this sentence in the Conclusions section and improved the wording throughout the manuscript for greater clarity where appropriate. [Lines 615-616, etc.]

Reviewer 2 Report

First of all, thank you so much for involving me in reviewing this manuscript.

Very interesting and current topic always of great study and debate especially in this period where the therapies are becoming more and more personalized.

Complex but well-conducted and understandable statistical analysis with well-structured graphs.

Clear and easily understood English language.

Adequate and recent bibliography.

Clear and understandable tables and images.

For me the article can be accepted in this form.

Author Response

Response to Reviewer 2

Comments and Suggestions for Authors

First of all, thank you so much for involving me in reviewing this manuscript.

Very interesting and current topic always of great study and debate especially in this period where the therapies are becoming more and more personalized.

Complex but well-conducted and understandable statistical analysis with well-structured graphs.

Clear and easily understood English language.

Adequate and recent bibliography.

Clear and understandable tables and images.

For me the article can be accepted in this form.

Response: We appreciate the reviewer’s positive comments.

Reviewer 3 Report

Using a genome-wide association study (GWAS ) using whole genome genotyping array in 350 patients undergoing laparoscopic-assisted colectomy, the authors of this article discuss the role of SNPs on different genes and their involvement in the various opioid sensitivity among individuals. The results may provide information for personalized pain management after laparoscopic-assisted colectomy.

I recommend reviewing citations and avoiding unnecessary self-quotations

I agree to accept it in its current form

Author Response

Response to Reviewer 3

Comments and Suggestions for Authors

Using a genome-wide association study (GWAS ) using whole genome genotyping array in 350 patients undergoing laparoscopic-assisted colectomy, the authors of this article discuss the role of SNPs on different genes and their involvement in the various opioid sensitivity among individuals. The results may provide information for personalized pain management after laparoscopic-assisted colectomy.

I recommend reviewing citations and avoiding unnecessary self-quotations

I agree to accept it in its current form

Response: We appreciate the reviewer’s positive comments. According to the reviewer’s suggestion, we reviewed the citations to avoid unnecessary self-quotations throughout the manuscript.
